# Variation Characteristics of Summer Water Vapor Budget and Its Relationship with the Precipitation over the Sichuan Basin



**Dongmei Qi** [1,2], **Yueqing Li** [1,2,*] **and Changyan Zhou** [1,2]

1   Institute of Plateau Meteorology, China Meteorology Administration, Chengdu 610072, China;
    qidongmei1983@163.com (D.Q.); zcy001124@163.com (C.Z.)
2   Heavy Rain and Drought-Flood Disasters in Plateau and Basin Key Laboratory of Sichuan Province,
    Chengdu 610072, China
*   Correspondence: yueqingli@163.com

**Abstract:** Based on the daily precipitation data from the meteorological stations in Sichuan and the monthly average ERA-Interim reanalysis data from 1979 to 2016, the variation characteristics of summer water vapor budget in the Sichuan Basin and its relationship with precipitation are discussed in this study. The results show that, in summer, the water vapor in the Sichuan Basin and its four sub-basins flows in from the southern and western boundaries and flows out through the eastern and northern boundaries, and the basin is obviously a water vapor sink. From 1979 to 2016, the water vapor inflow from the southern and western boundaries significantly decreased, as well as the water vapor outflow through the eastern boundary. The summer precipitation in the Sichuan Basin is significantly positively correlated with the water vapor inflow at the southern boundary and net water vapor budget of the basin in the same period, and it is negatively correlated with the water vapor outflow at the northern boundary. The southern and northern boundaries are the two most important boundaries for the summer precipitation in the Sichuan Basin. Additionally, this study reveals that, under the multi-scale topography on the east side of the Tibet Plateau, the spatio-temporal distribution of precipitation in the Sichuan Basin results from the interactions between the unique topography of the Sichuan Basin and the different modes of water-vapor transport from low latitudes. The atmospheric circulation over the key area of air–sea interaction in the tropical region and its accompanying systems, as well as the anomalies of regional circulations and water vapor transport over the eastern China and Sichuan Basin, are the main reasons for the variation in summer precipitation in the Sichuan Basin.

**Keywords:** Sichuan Basin; water vapor budget; summer precipitation; water resource variation

## 1. Introduction

Water resources are indispensable natural resources in production and human life as well as other social activities. Factors of climate change, ecological environment evolution and human activities, and climate change and its impact on water resources affect water resources and are important subjects worldwide [1,2]. In the past century, the characteristics of climate and environment have undergone significant changes, such as global warming, and climate change necessarily leads to changes in the hydrological cycle and the spatio-temporal redistribution of water resources [3,4]. Water vapor, as an important component of the atmosphere, absorbs solar radiation. It is an important greenhouse gas, which is closely related to precipitation and climate and plays a key role in the global water cycle and energy cycle [5]. Water vapor transport and budget are important components of the regional water balance, which are directly related to rainfall events and the climate near the ground [6–8]. Many researchers have studied the relationship of water vapor transport and budget with regional drought and flood [9–11]. Xu et al. [12,13] analyzed the water vapor source and sink of the Meiyu belt over the Yangtze River Basin and found that the integrated water vapor transport in drought and wet years in the Yangtze River Basin

shows an inverse feature. They have pointed out that the water vapor transport in the "large triangular fan pattern" region composed of the low-latitude activity source areas related to the Tibet Plateau, the South China Sea monsoon, and the Indian monsoon has an important impact on regional drought and flood in China. Zhou et al. [14] illustrated that, when the water vapor source is different from the climatology, the summer precipitation anomaly is induced. The increase in water vapor transport is one of the reasons for the increasing frequency of flood events from 1961 to 2005 [15]. Jiang et al. [16] found that the convergence and divergence of water vapor are closely related to precipitation over the Yangtze River Basin. Zhang et al. [17] studied the characteristics of water vapor transport in East Asia and its impact on drought and flood in North China. Ding et al. [18] referred that the long-lasting drought in North China is the result of prevailing continental warm high and the eastward movement of the water vapor conveyor belt. Simmonds et al. analyzed the summer water vapor transport and budget in China and found that the source of water vapor in southeastern China is from the South China Sea and the Bay of Bengal, while the water vapor in Northeast China and some parts of North China mainly comes from middle-latitude westerlies [19]. Yang et al. [20] compared the characteristics of water vapor transport for four summer rainfall patterns over the monsoon regions in the eastern China. Their study showed that the northern pattern is mainly affected by water vapor transport of Asian monsoon, the central pattern and Yangtze River pattern are affected by water vapor transport from the Pacific, and the South China pattern is impacted by water vapor transport from the Indian Ocean, the Pacific, and the South China Sea. Li et al. [21] studied two main water vapor conveyor belts that influence the summer drought and flood in eastern Southwest China. Chu et al. and Dong et al. [22–24] employed different models to analyze the water vapor source of the first rainy season in South China and the water vapor sources of midsummer precipitation in different regions of eastern China. They also revealed the relationship between water vapor transport over the Yangtze River basin in the Meiyu period and the key region in the Tibet Plateau. All the above studies have shown that water vapor transport and budget significantly influence regional drought and flood. Studies on the air water resource and its utilization are directly related to regional economic construction, human life, and economic development, so it is of great significance.

The Sichuan Basin is located in the east of the Tibet Plateau. Due to the special geographical location and complex topography, it is a sensitive, vulnerable, and key area for climate change. The water resource distribution and regional climate change there have distinct particularity and diversity. In summer, affected by the combined influence of various monsoon circulations, the cyclone made by the southern branch flow around the Tibet Plateau and the water vapor transport from the Pacific Ocean, the Bay of Bengal, and the Arabian Sea make the region rich in water vapor and precipitation. The atmospheric water vapor source, the track of water vapor transport, and the water vapor budget are the key links of regional water cycle in this region, especially the air water resources. They play a key role in the regional water balance and are closely related to the atmospheric circulation evolution and the variation of regional drought and flood [25–27]. Summer precipitation in the Sichuan Basin shows a decreasing trend, especially after the 1990s [28–30]. Zhou et al. [31] pointed out that the water vapor transported from the southern boundary of eastern Tibet Plateau and its surrounding regions decreases. Moreover, the decreasing trend is especially significant after the 1990s, and is not prone to precipitation in the Sichuan Basin. In the Sichuan Basin, the atmospheric forcing and circulation factors causing precipitation anomalies are complex, resulting in the great variations of inter-annual precipitation and its spatial distribution and the alternating occurrence of drought and flood [32]. Summer precipitation in the Sichuan Basin is affected by the simultaneity of East Asian summer monsoon, Tibetan Plateau summer monsoon, and southwest monsoon [33–36], as well as the anomalous variations of the location and intensity of the western Pacific subtropical high, South Asian high, and the westerly jet [37–41]. Rainfall events and climate in the basin are also closely related to variation in the atmospheric heat source in the Tibet Plateau and its surrounding regions [42–45]. In addition, as an important external forcing source

for the atmosphere ocean has a great impact on the anomalies of atmospheric circulation, and the intensity of western Pacific warm pool, and the sea surface temperature (SST) in the equatorial central-eastern Pacific and Indian Ocean is also closely related to the precipitation in the Sichuan Basin [46–50].

In summary, the anomalies of water vapor transport and budget are direct reasons for summer precipitation anomalies in China. Many researchers have studied the spatio-temporal distribution of summer precipitation in the Sichuan Basin. The analysis of the precipitation anomaly mainly focuses on the atmospheric circulation and external forcing signals such as SST, but few studies have systematically and deeply analyzed the water vapor transport and budget closely related to precipitation. Climate change in the Sichuan Basin, especially precipitation change, has significant regional characteristics. Consequently, it is necessary to investigate the precipitation characteristics and the specific response to climate change on a regional scale. It is enormously helpful to understand the characteristics and impacts of climate change and its regional response. The Sichuan Basin is a sensitive area for responding to climate change. Under global warming, the water vapor budget/transport in this region will change significantly, which will further result in abnormal changes of precipitation. In the past, research on the variation characteristics of water vapor budget in Sichuan mainly focused on a large spatial scale, and there were no in-depth discussions on each region within the basin. Then, what are the spatio-temporal variation characteristics of water vapor budget over the Sichuan Basin on a regional scale? In particular, what are the similarities and differences of water vapor budget changes among different regions in the basin? Is there is a close relationship between the water vapor budget/transport and regional droughts/floods, and how does the water vapor budget/transport over the Sichuan Basin influence summer precipitation there? What are the main factors for the water vapor variations and the mechanisms? At present, the possible answers to these questions are not clear. Focusing on these problems, this study employs the ERA-Interim reanalysis data to study the variation characteristics of the summer water vapor budget in the Sichuan Basin and uses the observed precipitation data at meteorological stations to reveal the impact of regional water vapor budget on the summer precipitation in the Sichuan Basin. It can improve the understanding of water vapor budget anomalies and its impact on the summer precipitation, and can support for the forecast of climatic disasters.

## 2. Material and Methods

The daily rain gauge data from the latest version (V3) of surface climatological data is compiled by the China National Meteorological Information Center. Out of the daily precipitation data at 114 stations in Sichuan Province from 1979 to 2016, 74 stations are selected to represent the Sichuan Basin. In addition, we use the monthly mean ERA-Interim reanalysis data with a horizontal resolution of $1.5° \times 1.5°$ from 1979 to 2016, including the specific humidity from 1000 hPa to 300 hPa, horizontal zonal wind, horizontal meridional wind, geo-potential height, and the corresponding surface pressure.

As shown in Figure 1, the boundary shows the Sichuan province, and the rectangles outline the Sichuan Basin. The study area is the rectangular area of the Sichuan Basin (Figure 1), in which the Sichuan Basin has six boundaries. Of these, 1 and 5 are the southern boundaries; 2 is the western boundary; 3 is the northern boundary; and 4 and 6 are the eastern boundaries (Figure 1a). The water vapor transport through the southern boundary of the basin is the sum of the water vapor transport at boundaries 1 and 5, and the water vapor transport through the eastern boundary is the sum of the water vapor transport at boundaries 4 and 6. Bounded by 105° E, the Sichuan Basin is divided into the eastern basin and the western basin (Figure 1b). Additionally, boundaries 1, 2, 3, and 4 are defined as the southern, western, northern, and eastern boundaries of the western basin, respectively; boundaries 9 and 7 are the southern boundaries of the eastern basin; boundary 4 is the western boundary of the eastern basin; boundary 5 is the northern boundary of the eastern basin; and boundaries 6 and 8 are defined as the eastern boundaries of the eastern basin.

The water vapor transport through the southern boundary of the eastern basin is the sum of the water vapor transport at boundary 9 and 7. The water vapor transport through the eastern boundary of the eastern basin is the sum of the water vapor transport at boundary 6 and 8. Along 30° N, the Sichuan Basin is divided into the northern basin and the southern basin (Figure 1c), and both regions have four refined boundaries. Boundaries 1, 2, 3, and 4 are the southern, western, northern, and eastern boundaries of the southern basin; and boundaries 5, 6, 7, and 8 are the southern, western, northern, and eastern boundaries of the northern basin. In this study, the total inflow and outflow over the Sichuan Basin (the eastern, western, northern, and southern basin) are the sum of the inflow and outflow at the southern, western, northern, and eastern boundaries, and the difference between the total inflow and outflow is the net budget. The net water vapor budget over the Sichuan Basin is defined as $B_T$, including the water vapor budgets at the southern ($B_S$), western ($B_W$), northern ($B_N$), and eastern ($B_E$) boundaries. Here, S, W, N, and E, respectively, stands for the southern, western, northern, and eastern boundary. The positive (negative) values of $B_S$, $B_W$, $B_N$, and $B_E$ indicate water vapor inflow (outflow) on the boundary of the region. The variations in summer water vapor inflow and outflow over the Sichuan Basin and its eastern, western, northern and southern sub-basins, and their relationship with precipitation are analyzed in the following.

The Equations to calculate the vertically integrated water vapor flux are as follows.

The zonal and meridional water vapor fluxes are calculated as follows:

$$Q_\lambda = -\frac{1}{g} \int_{P_s}^{P_t} qu \, dp \tag{1}$$

$$Q_\varphi = -\frac{1}{g} \int_{P_s}^{P_t} qv \, dp \tag{2}$$

where $P_s$ is the surface pressure, and $P_t$ is set as 300 hPa; $q$ is the specific humidity, $u$ is the meridional wind, $v$ is the zonal wind, $\lambda$ is longitude, $\varphi$ is latitude, and $g$ is the gravity acceleration. The water vapor transport calculated by the boundary integral is as follows:

$$F_u = \int Q_\lambda a \, d\varphi \tag{3}$$

$$F_v = \int Q_\varphi a \cos \varphi \, d\lambda \tag{4}$$

In this Equation, $F_u$ is the east–west water vapor transport, and $F_v$ is the north–south water vapor transport.

The regional total water vapor budget is

$$D_s = \sum (F_u, F_v) = F_i - F_a \tag{5}$$

In this Equation, $F_i$ is the total water vapor inflow, and $F_a$ is the total water vapor outflow.

Correlation and composite analyses are utilized in detecting the relationships between pairs of variables. The change trend is analyzed by linear trend estimation method and Student's t test is used to assess the statistical significance. The summer is the averages of June, July, and August.

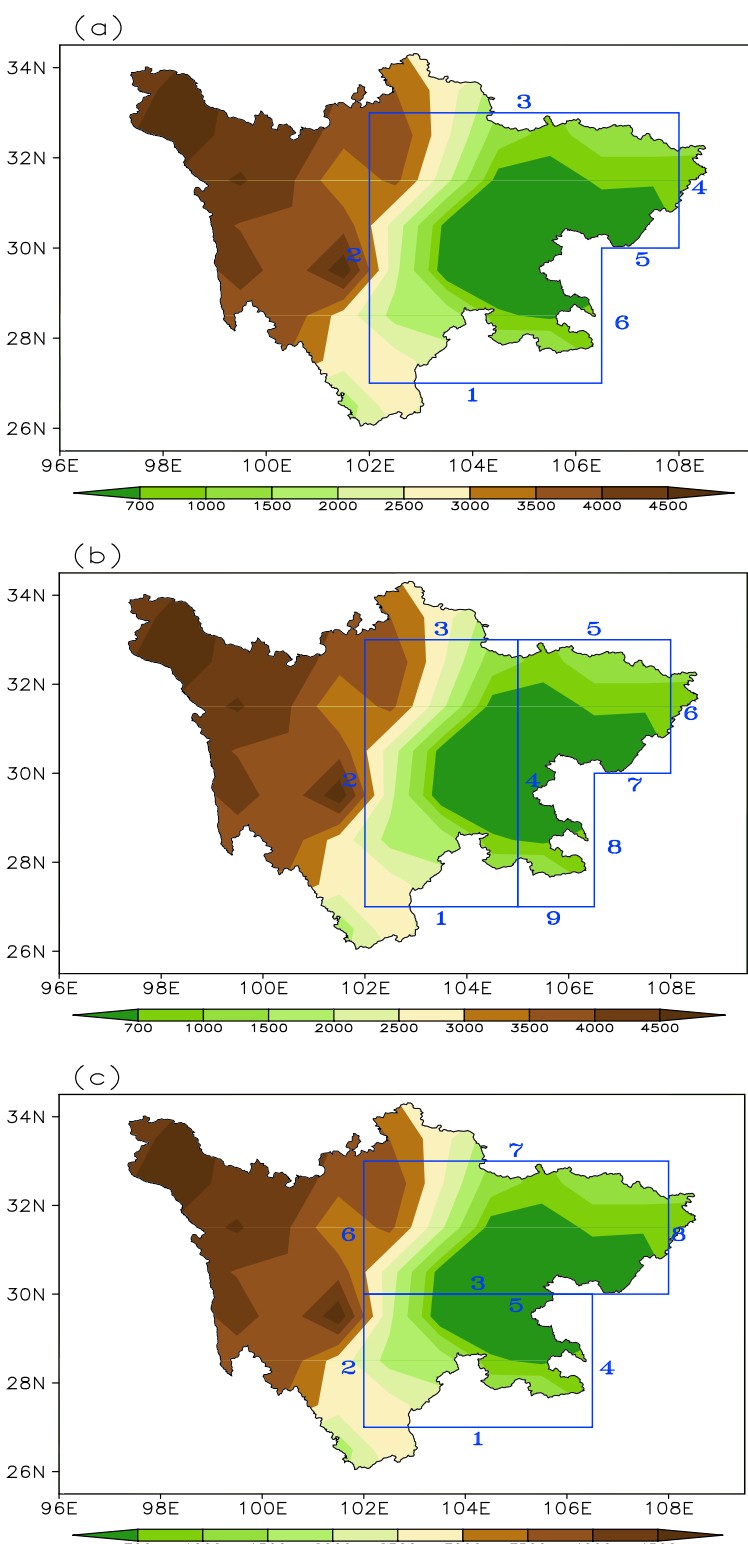

**Figure 1.** The boundaries of the Sichuan Basin and the sub-basins. (Map shows the topography of Sichuan Province, different colors indicate terrain height, units: m. The boundary shows the Sichuan province, and the rectangles outline the Sichuan Basin. (**a**) The Sichuan Basin; (**b**) eastern and western basin; (**c**) northern and southern basin).

## 3. Results

### 3.1. Variation Characteristics of the Summer Water Vapor Budget in the Sichuan Basin

As can be seen in Table 1, the summer water vapor flows in from the southern and western boundaries and flows out through the eastern and northern boundaries of the Sichuan Basin and the four sub-basins. The amount of outflow through the eastern boundary is much greater than that through the northern boundary, and thus the eastern boundary is the main boundary of the water vapor outflow over the Sichuan Basin. The amount of water vapor inflow from the southern boundary is much greater than that from the western boundary for all the sub-basins in the Sichuan Basin, except for the western basin. The total inflow is much greater than the total outflow at the boundaries of the Sichuan Basin and the four sub-basins, and the net water vapor budget is positive. Therefore, an obvious water vapor sink is observed in the Sichuan Basin and the four sub-basins in summer. Comparing the eastern and western basin, except for the inflow from the western boundary, the water vapor inflow and outflow at each boundary, the total inflow, the total outflow, and the net budget in the eastern basin are larger than those in the western basin, indicating that the variation in water vapor budget over the eastern basin is more significant than that over the western basin. This is closely related to the unique topography of high and steep mountain ranges on the boundary of the western basin and the relatively low and gentle mountain ranges on the edge of eastern basin, and their different interactions with the water vapor from the peripheral circulation. In comparing the northern and southern basin, the water vapor inflow at the western and southern boundaries, and the water vapor outflow at the eastern and northern boundaries, the total inflow and the total outflow are all greater in the southern basin than those in the northern basin, while the net water vapor budget over the two sub-basins is quite close.

**Table 1.** The net water vapor transport, total inflow, total outflow, and the net budget at all boundaries in the Sichuan Basin (the eastern, western, northern and southern basin) in summer (unit: $10^6$ kg s$^{-1}$).

| Region | Water Vapor Budget | | | | | | |
|---|---|---|---|---|---|---|---|
| | $B_S$ | $B_W$ | $B_N$ | $B_E$ | Inflow | Outflow | $B_T$ |
| Basin | 106.76 | 59.79 | −24.09 | −67.63 | 166.55 | −91.72 | 74.83 |
| eastern | 69.06 | 55.55 | −20.32 | −67.63 | 124.61 | −87.95 | 36.66 |
| western | 33.88 | 59.79 | −3.32 | −55.55 | 93.67 | −58.87 | 34.8 |
| northern | 68.22 | 28.22 | −24.09 | −31.01 | 96.44 | −55.1 | 41.34 |
| southern | 70.74 | 30.6 | −30.85 | −36.62 | 101.34 | −67.47 | 33.87 |

From the inter-annual variation in the water vapor budget at all boundaries of the Sichuan Basin in summer (Figure 2a), it is found that the water vapor inflow from the southern and western boundaries of the Sichuan Basin decreases significantly from 1979 to 2016 (significant at the 90% and 95% confidence level, respectively), and the water vapor outflow through the northern boundary increases (but not significant at the 90% confidence level), while the water vapor outflow through the eastern boundary decreases significantly (significant at the 95% confidence level). Moreover, the net budget of the water vapor over the whole basin also decreases in summer (but not significant at the 90% confidence level). The decrease in water vapor inflow at the western and southern boundaries is mostly balanced with the decrease in water vapor outflow at the eastern boundary, while the increase in water vapor outflow at the northern boundary is the main reason for the decrease in net water vapor budget in the Sichuan Basin. From 1979 to 2016, the variation trends of water vapor at all the boundaries of the eastern and western Sichuan Basin are the same (Figure 2b,c), and the water vapor inflow at their southern and western boundaries shows a decreasing trend (significant at the 90% and 95% confidence level, respectively). In both regions, the decline at the western boundary is greater than that at the southern boundary. The water vapor outflow at the northern boundary shows a weak increasing

trend (but not significant at the 90% confidence level), while the water vapor outflow at the eastern boundary obviously decreases (significant at the 95% confidence level). The net water vapor budget in the eastern and western basin shows a decreasing trend (but not significant at the 90% confidence level), and the decrease in net water vapor budget in the western basin is more obvious than that in the eastern basin in summer. In comparison, the decline in water vapor inflow (outflow) at each boundary in the western basin is weaker than that in the eastern basin, but the decline in net water vapor budget in the western basin is greater than that in the eastern basin. In fact, the decrease in water vapor inflow at the western and southern boundaries is largely balanced with the decrease in water vapor outflow at the eastern boundary, and the increase in water vapor outflow at the northern boundary is the main reason for the decrease in net water vapor budget in the eastern and western basin. In the summers of 1979–2016, the water vapor inflow at the southern boundary exhibits a decreasing trend (but not significant at the 90% confidence level). Meanwhile, the water vapor inflow at the western boundary exhibits a significant decreasing trend in the northern basin (significant at the 99% confidence level), and the decrease in water vapor inflow at the western boundary is larger than that at the southern boundary. The water vapor outflow shows an increasing trend at the northern boundary (but not significant at the 90% confidence level) and an obviously decreasing trend at the eastern boundary (significant at the 99% confidence level) (Figure 2d). For the southern basin (Figure 2e), the water vapor inflow at the southern boundary shows a decreasing trend (significant at the 90% confidence level), and the inflow at the western boundary also shows a decreasing trend (but not significant at the 90% confidence level). The decrease in water vapor inflow at the southern boundary is larger than that at the western boundary. The water vapor outflow at the northern and eastern boundaries shows a decreasing trend (but not significant at the 90% confidence level), and the water vapor outflow at the eastern boundary decreases more strongly than that at the northern boundary. It is worth noting that the decline in water vapor inflow at the western boundary of the northern basin is greater than that in the southern basin; the decline in water vapor inflow at the southern boundary is weaker than that in the southern basin, and the decline in water vapor outflow at the eastern boundary of the northern basin is significantly greater than that in the southern basin. In summer, the net water vapor budget increases slightly in the northern basin (but not significant at the 90% confidence level) but decreases in the southern basin (significant at the 90% confidence level).

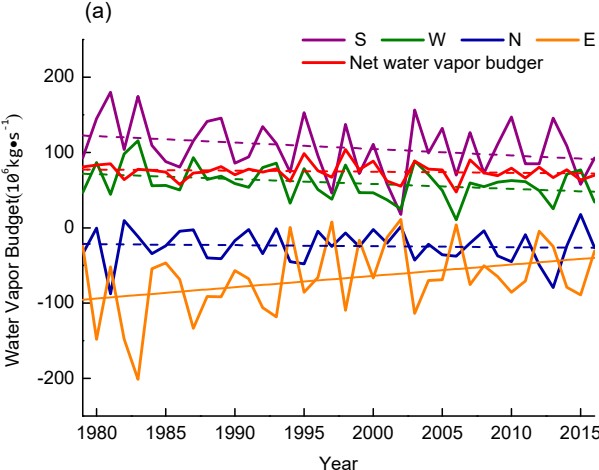

**Figure 2.** *Cont.*

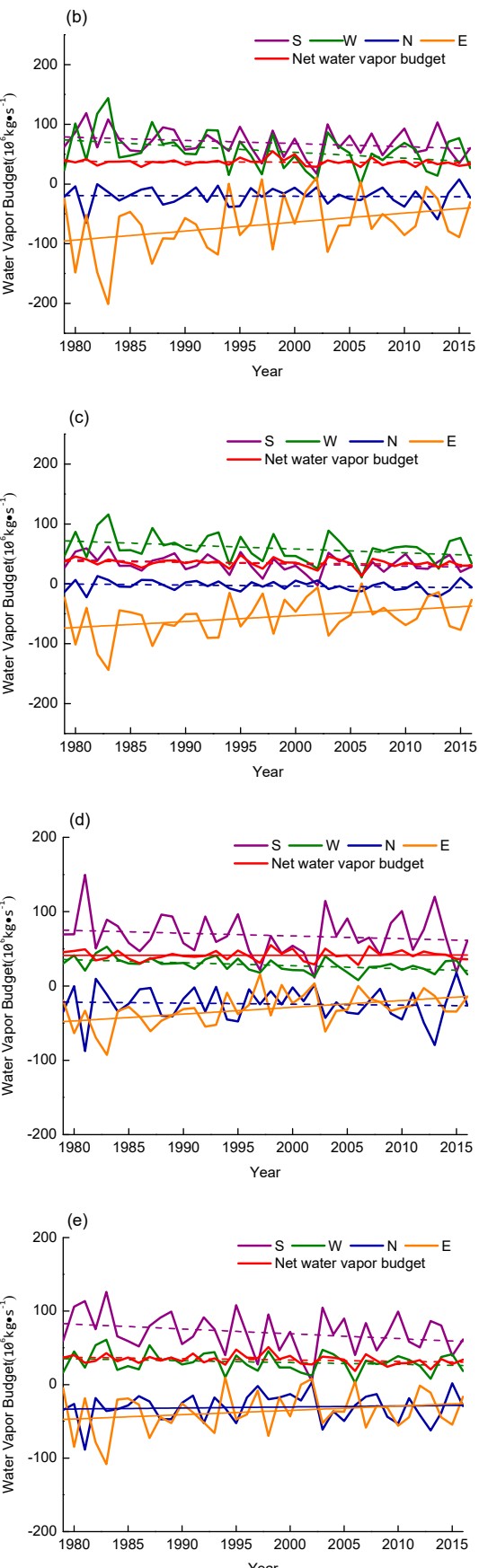

**Figure 2.** Inter-annual variation of water vapor inflow and outflow and net water vapor budget at

each boundary of the Sichuan Basin in summer ((**a**) the Sichuan Basin, (**b**) eastern basin, (**c**) western basin, (**d**) northern basin, and (**e**) southern basin; S, W, N, and E indicate the southern, western, northern, and eastern boundaries, respectively; unit: $10^6$ kg s$^{-1}$. The thin lines indicate linear trends, the solid and dashed lines show positive and negative trends, respectively).

### 3.2. Spatio-Temporal Variation of Summer Precipitation in the Sichuan Basin

Sichuan is located in the subtropical climate zone. It is mainly composed of the Western Sichuan Plateau in the west and the Sichuan Basin in the east. The meteorological and geological conditions are complex and diverse, with high incidence of natural disasters, resulting in serious losses and wide impact. Due to the special geographical location and significantly different topography, as well as the alternating influence of different monsoon circulations, a unique regional climate type is formed, with significantly different and various variations of precipitation, and the causes are complex. As shown in Figure 3, the summer precipitation in Sichuan province is mainly over the Sichuan Basin, and there is more precipitation in the western basin than in the eastern basin, while there is less precipitation in the Western Sichuan Plateau. Moreover, there are three centers of precipitation maxima in the Sichuan Basin, which are located in the northwest of the basin (centered around Beichuan) with the summer precipitation of 765 mm, in the northeast of the basin (centered around Wanyuan) with the summer precipitation of 672 mm and in the southwest of the basin (centered around Ya'an) with the summer precipitation of 960 mm.

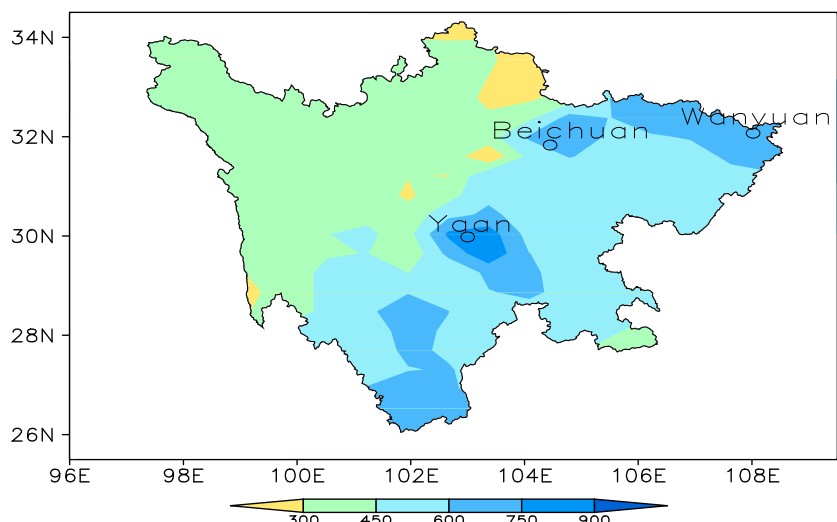

**Figure 3.** Spatial distribution of summer mean precipitation in the Sichuan from 1979 to 2016 (shadow; unit, mm; black circles are representative stations of the three precipitation centers in the basin).

In addition, the southern margin of the Western Sichuan Plateau is also a relative center of precipitation maxima. The locations of the three precipitation centers are closely related to the topography. Ya'an, in the southwestern basin, has the largest precipitation, known as the "sky leakage", followed by Beichuan in the northwestern basin and Wanyuan in the northeastern basin. The large precipitation is mainly distributed along the southwest, northwest, and northeast transition areas at the junction of the Sichuan Basin and its surrounding mountains in the western and northern basin.

Figure 4 shows the temporal variation of precipitation at the representative stations of the three precipitation centers in the Sichuan Basin from 1979 to 2016. The summer precipitation in the Sichuan Basin shows a decreasing trend from 1979 to 2016, with a decrease of 18.87 mm 10a$^{-1}$ (significant at the 90% confidence level); the summer precipitation in Beichuan in the northwestern basin shows a decreasing trend, with a decrease of 50.25 mm·10a$^{-1}$; the summer precipitation in Wanyuan in the northeastern basin also displays a decreasing trend, with a decrease of 36.57 mm 10a$^{-1}$. The summer precipi-

tation in Ya'an in the southwestern basin presents a decreasing trend, with a decrease of 21.42 mm $10a^{-1}$ (but not significant at the 90% confidence level). In comparison, the decreasing trend of summer precipitation is the most obvious in Beichuan, followed by Wanyuan, and it is the weakest in Ya'an.

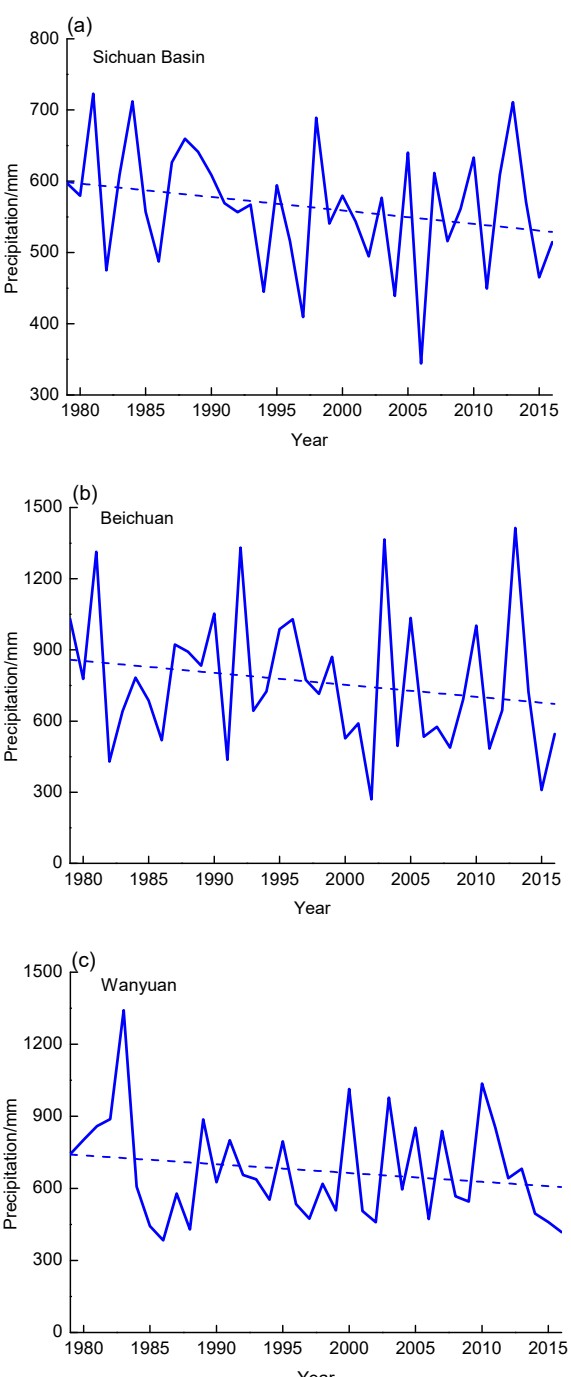

**Figure 4.** *Cont.*

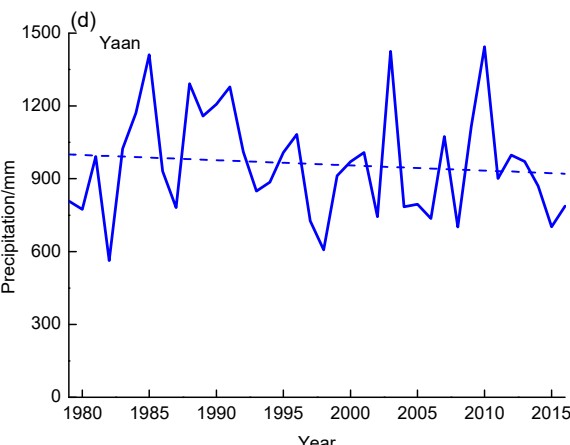

**Figure 4.** Inter-annual variation of summer precipitation at the three representative stations with large precipitation in the Sichuan Basin from 1979 to 2016. ((**a**) the Sichuan Basin, (**b**) Beichuan, (**c**) Wanyuan, and (**d**) Ya'an. The thin lines indicate linear trends.).

*3.3. Relationship between the Summer Precipitation and Water Vapor Budget in the Sichuan Basin*

3.3.1. Correlation Analysis

As shown in Table 2, the summer precipitation in the Sichuan Basin is significantly positively correlated with the water vapor inflow at the southern boundary of the basin (eastern, western, northern, and southern basin) and the net water vapor budget in the same period, and negatively correlated with the water vapor outflow at the northern and eastern boundaries of the basin (the eastern and northern basin) in the same period. In addition, the water vapor inflow from the southern boundary plays a key role for the summer precipitation in the whole basin and its sub-basins, followed by the water vapor outflow at the northern boundary. The water vapor outflow through the northern boundary makes a more important impact in the eastern basin than that in the western basin. The net water vapor budget in the whole basin as well as the eastern, western, and northern basin has a great impact on the summer precipitation over the basin. The summer precipitation in Beichuan has a significant positive correlation with the water vapor budget at the southern boundary of the basin (four sub-basins) in the same period and a significant negative correlation with the water vapor budget at the northern boundary. Except for the northern basin, the influence of water vapor budget at the northern boundary is greater than that at the southern boundary. Moreover, it is also positively correlated with the net water vapor budget of the basin (the eastern, western, and northern basin) in the same period. The summer precipitation in Ya'an has a positive correlation with the water vapor budget at the southern boundary of the basin (four sub-basins) and a negative correlation with the water vapor budget at the northern boundary (the eastern, northern, and southern basin). The influence of the water vapor budget over the eastern basin on the precipitation in Ya'an is greater than that over the western basin, the influence of the water vapor budget at the southern boundary of the northern basin is greater than that at the northern boundary, and the influence of water vapor budget at the northern boundary of the southern basin is greater than that at the southern boundary. The summer precipitation in Wanyuan has a significant positive correlation with the water vapor budget and net water vapor budget at the southern and western boundaries of the basin (four sub-basins) in the same period, a significant negative correlation with the water vapor budget at the eastern boundary, and a negative correlation with the water vapor budget at the northern boundary of the southern basin. In comparison, the influence of the net water vapor budget in the western basin is greater than that in the eastern basin, and the influence of the net water vapor budget in the northern basin is greater than that in the southern basin. The influence of the water vapor budget at each boundary is greater in the southern basin than that in the northern basin.

**Table 2.** Correlation Coefficients of the summer precipitation in the Sichuan Basin and at the three precipitation centers with the water vapor budget in the same period from 1979 to 2016. (**** represents passing the 99.9% confidence level, *** represents passing the 99% confidence level, ** represents passing the 95% confidence level, and * represents passing the 90% confidence level).

| Water Vapor Budget | | $R_{Sichuan\ Basin}$ | $R_{Beichuan}$ | $R_{Yaan}$ | $R_{Wanyuan}$ |
|---|---|---|---|---|---|
| Sichuan basin | $B_S$ | 0.686 **** | 0.601 **** | 0.349 ** | 0.654 **** |
| | $B_W$ | 0.235 | 0.056 | 0.037 | 0.471 *** |
| | $B_N$ | −0.423 *** | −0.684 **** | −0.369 ** | −0.111 |
| | $B_E$ | −0.291 * | −0.081 | −0.057 | −0.56 **** |
| | $B_T$ | 0.614 **** | 0.34 ** | 0.22 | 0.468 *** |
| eastern basin | $B_S$ | 0.715 **** | 0.635 **** | 0.38 ** | 0.629 **** |
| | $B_W$ | 0.227 | 0.055 | 0.034 | 0.511 **** |
| | $B_N$ | −0.412 *** | −0.68 **** | −0.395 ** | −0.12 |
| | $B_E$ | −0.291 * | −0.081 | −0.057 | −0.56 **** |
| | $B_T$ | 0.566 **** | 0.295 * | 0.143 | 0.377 ** |
| western basin | $B_S$ | 0.599 **** | 0.518 **** | 0.29 * | 0.63 **** |
| | $B_W$ | 0.235 | 0.056 | 0.037 | 0.471 *** |
| | $B_N$ | −0.311 * | −0.582 **** | −0.246 | 0.002 |
| | $B_E$ | −0.227 | −0.055 | −0.034 | −0.511 **** |
| | $B_T$ | 0.587 **** | 0.3 * | 0.27 | 0.488 *** |
| northern basin | $B_S$ | 0.666 **** | 0.747 **** | 0.435 *** | 0.468 *** |
| | $B_W$ | 0.232 | 0.063 | 0.053 | 0.377 ** |
| | $B_N$ | −0.423 *** | −0.684 **** | −0.369 ** | −0.111 |
| | $B_E$ | −0.313 * | −0.16 | −0.122 | −0.503 *** |
| | $B_T$ | 0.672 **** | 0.391 ** | 0.257 | 0.469 *** |
| southern basin | $B_S$ | 0.631 **** | 0.537 **** | 0.307 * | 0.671 **** |
| | $B_W$ | 0.205 | 0.027 | 0.012 | 0.49 *** |
| | $B_N$ | −0.556 **** | −0.752 **** | −0.421 *** | −0.349 ** |
| | $B_E$ | −0.253 | −0.012 | −0.002 | −0.567 **** |
| | $B_T$ | 0.302 * | −0.01 | 0.034 | 0.346 ** |

From the above demonstration, the impact of the water vapor budget at the southern boundary on the summer precipitation in the Sichuan Basin is the greatest, followed by the northern boundary, and it is also closely related to the regional net water vapor budget. For the precipitation in Beichuan, in the northern basin, the water vapor budget at the southern boundary makes influence on it most significantly, followed by the northern boundary, while the influence of the water vapor budget in other sub-basins are all the most significant at the northern boundaries. For the precipitation in Ya'an, the influence of water vapor budget is the greatest at the northern boundaries of the eastern, southern, and whole basin and at the southern boundaries of the western and northern basin. For the precipitation in Wanyuan, the impact of water vapor budget at the eastern boundary is the greatest in the northern basin, while in the whole basin and the other parts of the basin, the impact is the greatest at the southern boundaries.

### 3.3.2. Composite Analysis

Five wet summers (summer precipitation is greater than 650 mm; 1981, 1984, 1988, 1998, and 2013) and five drought summers (summer precipitation is less than 450 mm; 1994, 1997, 2004, 2006, and 2011) in the Sichuan Basin are selected for composite analysis to discuss the differences in summer water vapor inflow, outflow and budget in the Sichuan Basin between the wet and drought years.

As seen in Table 3, the water vapor inflow at the southern and western boundaries, the water vapor outflow at the northern and eastern boundaries, and the total inflow, total outflow, and net water vapor budget in the wet summers in the Sichuan Basin are much greater than those in the drought summers. Between the wet summers and drought summers, for the whole basin, the difference in the water vapor inflow is the largest at

the southern boundary, followed by the water vapor outflow at the eastern and northern boundaries, and the difference in the water vapor inflow at the western boundary is the least; for the four sub-basins, the difference in the water vapor inflow is the greatest at the southern boundary. The difference in the water vapor outflow is the least at the northern boundaries of the eastern and western basin and at the western boundaries of the northern and southern basin. Out of the four sub-basins, the difference in water vapor inflow between the wet and drought summers at the southern boundary is the greatest in the northern basin, followed by that in the southern and eastern basin, and that in the western basin is the least.

**Table 3.** The water vapor flux, total inflow, total outflow, and net budget at each boundary of the Sichuan Basin in the wet and drought summers (unit: $10^6$ kg s$^{-1}$).

| Region | Summers | Water Vapor Budget Region | | | | | | |
|---|---|---|---|---|---|---|---|---|
| | | $B_S$ | $B_W$ | $B_N$ | $B_E$ | Inflow | Outflow | $B_T$ |
| Sichuan | wet | 713.4 | 272.7 | −248.4 | −331.5 | 986.1 | −579.9 | 406.2 |
| basin | drought | 374.4 | 213.3 | −138.3 | −129 | 587.7 | −267.3 | 320.4 |
| eastern | wet | 480.9 | 248.4 | −194.7 | −331.5 | 729.3 | −526.2 | 203.1 |
| basin | drought | 254.7 | 152.4 | −116.7 | −129 | 407.1 | −245.7 | 161.4 |
| western | wet | 211.5 | 272.7 | −47.4 | −248.4 | 484.2 | −295.8 | 188.4 |
| basin | drought | 99.9 | 213.3 | −25.5 | −152.4 | 313.2 | −177.9 | 135.3 |
| northern | wet | 513.3 | 136.2 | −248.4 | −167.1 | 649.5 | −415.5 | 234 |
| basin | drought | 262.2 | 103.8 | −138.3 | −53.1 | 366 | −191.4 | 174.6 |
| southern | wet | 452.4 | 129.6 | −251.7 | −164.4 | 582 | −416.1 | 165.9 |
| basin | drought | 234 | 106.2 | −117.3 | −75.9 | 340.2 | −193.2 | 147 |

The water vapor transport, especially the water vapor inflow at the southern boundary, plays a dominant role in the summer precipitation over the Sichuan Basin. To a great extent, the amount of water vapor flux determines whether it is a wet year or a drought year in the Sichuan Basin, especially the water vapor inflow at the southern boundary of the northern basin.

By analyzing the water vapor flux, the source and path of water vapor over the Sichuan Basin can be learned. For the wet summers in the Sichuan Basin (Figure 5a), over the sea east of Philippines and north of East Indonesia–Papua New Guinea (120–180° E, 0–10° N), there are consistent easterly airflows, the area (120–170° E, 10–20° N) is dominated by an anticyclone, the area (130–160° E, 20–35° N) is dominated by an obvious cyclone, and the area (60–85° E, 5°–30° N) is dominated by a weak cyclone. Over southern and eastern China, there is an anomalous divergence of water vapor. The vast region east of the Tibet Plateau, including the Sichuan Basin, is dominated by southerly airflows. Due to the terrain of the nearly north–south steep slope east of the Tibet Plateau and the block of nearly east–west mountains in the north of the basin, the anomalous water vapor convergence extends from the whole Sichuan Basin to the south of the Western Sichuan Plateau, which is conducive to precipitation in the Sichuan Basin. On the contrary, in the drought summers (Figure 5b), over the sea (120–180° E, 0–10° N), there are consistent westerly airflows, the area (120–160° E, 10–25° N) is dominated by a cyclone, and the area (60–90° E, 5° S–20° N) is dominated by an anticyclone. There are convergence anomalies of water vapor in southern and eastern China. The vast region east of the Tibet Plateau, including the Sichuan Basin, is dominated by easterly airflows. There is an anomalous water vapor divergence over the Sichuan Basin and its surroundings, leading to less precipitation in the basin. It shows that the anomalous southerly (easterly) airflow and water vapor convergence (divergence) persist in wet (drought) summers in the Sichuan Basin, and the anomalous water vapor divergence (convergence) is maintained in southern and eastern China.

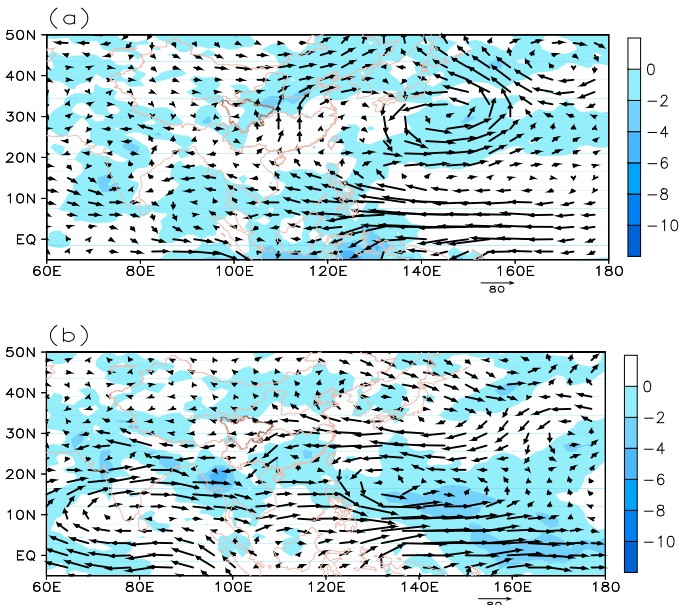

**Figure 5.** Anomalies of integrated water vapor flux in the whole layer (vector; unit: kg m$^{-1}$ s$^{-1}$) and the water vapor divergence (shaded; unit: $10^{-5}$ kg m$^{-2}$ s$^{-1}$) in (**a**) wet summers and (**b**) drought summers in the Sichuan Basin.

Moreover, in the drought summers, the water vapor transport and convergence in eastern and southern China are strong and link to the strong water vapor convergence over the South China Sea at low latitudes. In the wet summers, the water vapor divergence is located south of the basin at low latitudes, which is broken from the water vapor convergence over the South China Sea. Therefore, the amount of summer precipitation in the Sichuan Basin is closely related to the circulation anomalies over the key sea area east of Philippines and north of East Indonesia–Papua New Guinea (120–180° E, 0–10° N), and the circulation anomalies to the north of the key area. When there is a consistent easterly airflow in the key sea area and an anticyclonic circulation to its north (120–170° E, 10–20° N), the corresponding descending motion over it is enhanced. The significant easterly wind anomaly over the equatorial western Pacific extends westward, which helps the warm and humid airflow at low latitudes travel to the Sichuan Basin and increases the water vapor inflow at the southern boundary of the Sichuan Basin. As the Sichuan Basin is located in the convergence zone of water vapor, it favors summer precipitation, and vice versa.

The summer precipitation in the Sichuan Basin is closely related to the anomalies of the regional water vapor budget and convergence. The atmospheric circulation anomaly is the direct reason for the anomalies of water vapor and precipitation in summer over the Sichuan Basin. What is the circulation anomaly that causes the anomalies of water vapor and precipitation in summer over the Sichuan Basin? We analyze the characteristics of the atmospheric circulation in the wet and drought summers in the following. From the difference in composite circulation fields between the wet summers and drought summers in the Sichuan Basin, it is shown that there is more water vapor transported by the easterly wind (Figure 6a) near the equator at 700 hPa, which causes the water vapor to continuously travel westward from the South China Sea and the western Pacific regions. Part of the warm and humid air moves along the western edge of subtropical high and turns into southwesterly wind to transport the water vapor to the Sichuan Basin. Another part transports to the west and converges with the warm and humid flow in the Bay of Bengal. Influenced by the anomalous anticyclone above the area from the south of the plateau to the Bay of Bengal, part of the water vapor turns around and follows the anticyclonic circulation to transports along the southern edge of the plateau into the Sichuan Basin. As for the 850 hPa geo-potential height (Figure 6b), the low pressure trough

east of Baikal Lake is obviously strong. Over the sea east of Philippines and north of East Indonesia–Papua New Guinea (135–175° E, 10–20° N), there is a significantly positive anomaly, which indicates that, when the anticyclonic circulation over the ocean east of Philippines is anomalously strong, the corresponding descending motion over the region is enhanced. A significant easterly wind anomaly exists over the equatorial western Pacific region and extends westward, which helps the humid and warm flow at low latitudes transport to the Sichuan Basin, leading to the increase in water vapor inflow at the southern boundary and the increase in summer precipitation in the Sichuan Basin. It is consistent with the water vapor flux shown in Figure 5. The configuration of 700 hPa wind field and 850 hPa geo-potential height field is the main reason for the variations in water vapor budget and precipitation in summer in the Sichuan Basin.

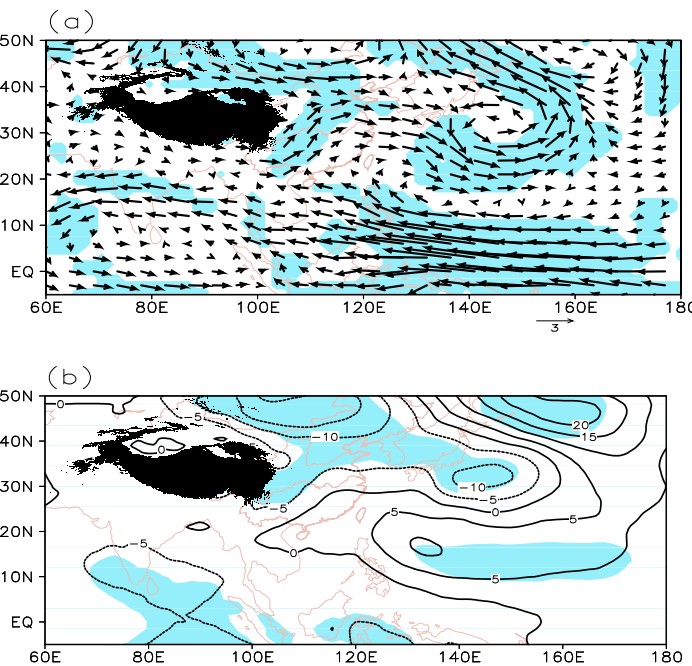

**Figure 6.** The difference of the composites of (**a**) 700 hPa wind (unit: m s$^{-1}$) and (**b**) 850 hPa geopotential height (unit: gpm) between wet summers and drought summers in the Sichuan Basin. The blue shadow indicates passing the 95% confidence level, and the black shadow indicates the Tibet Plateau.

## 4. Conclusions

Using many statistical and diagnostic methods, this study employed the daily precipitation data from the meteorological stations in Sichuan, and the monthly averaged ERA-Interim reanalysis data to study the variation characteristics of summer water vapor budget and its relationship with the precipitation in the Sichuan Basin. The main conclusions are as follows.

The summer water vapor flows in from the southern and western boundaries and flows out at the eastern and northern boundaries in the Sichuan Basin and the four sub-basins. As the main boundary of water vapor outflow, the amount of water vapor outflow at the eastern boundary is much larger than that at the northern boundary. Except for the western basin, where the water vapor inflow at the western boundary is the largest, the water vapor inflow is the largest at the southern boundaries in the other sub-basins. From 1979 to 2016, the water vapor inflow from the southern and the western boundaries of the Sichuan Basin and its sub-basins to Sichuan Province decreased significantly, as well as the water vapor outflow through the eastern boundary. Except for the southern basin, all the outflows through the northern boundaries increased; except for the northern basin, the net water vapor budget showed a decreasing trend in summer.

There are three large-value centers of the summer precipitation over the Sichuan Basin located in Beihuan, Wanyuan, and Ya'an, respectively, and Ya'an has the largest precipitation. The summer precipitation shows a significantly positive correlation with both the water vapor inflow from the southern boundary and the net budget in the Sichuan Basin and a negative correlation with water vapor outflow at the northern boundary in the same period. For the whole Sichuan Basin and its four sub-basins, the southern and northern boundaries are the most important boundaries for summer precipitation. Relatively, the influence of water vapor budget at all the boundaries in the eastern basin on the precipitation is greater than that in the western basin, while the influence is greater in the northern basin than that in the southern basin. By comparing the three centers with large precipitation values, it is found that the influence of outflow at the northern boundaries on the summer precipitation in Beichuan and Ya'an is the greatest, and the inflow at the southern boundary impacts the precipitation in Wanyuan the most greatly.

In the wet summers of the Sichuan Basin, the water vapor inflow from the southern and western boundaries, the total water vapor inflow, the outflow at the northern and eastern boundaries, and the total water vapor outflow are significantly larger than those in the drought summers, as well as the net budget. The difference in water vapor inflow between the wet summers and drought summers are greatest at the southern boundary of the northern basin. The water vapor flux in summer, especially the water vapor inflow at the southern boundary, plays a dominant role in summer precipitation over the basin. To some extent, the summer water vapor in the Sichuan Basin, especially the water vapor inflow from the southern boundaries of the northern basin, decides whether the summer is wet or dry. The impact of water vapor inflow from the southern boundary of the northern basin plays a particularly significant role.

The summer precipitation in the Sichuan Basin is not only directly related to the regional circulation and water vapor transport in eastern China and the Sichuan Basin, but also closely related to the atmospheric circulation over the key area of air–sea interaction in the tropics. It reveals that, under the complex topography on the east side of the Tibet Plateau, the spatio-temporal distribution of precipitation in the Sichuan Basin is the result of the new mechanism of multi-scale interactions between the unique topography of the Sichuan Basin and the different water vapor transports at low latitudes. When there are easterly (westerly) airflows east of the Philippines and north of East Indonesia–Papua New Guinea (120–180° E, 0–10° N) and an anticyclonic (cyclonic) circulation in the north, the anomalous southerly (easterly) airflow and water vapor divergence (convergence) maintain in the key region of the eastern and southern China. Meanwhile, due to the complex topography of the plateau and basin, the anomalies of regional water vapor budget and divergence in the Sichuan Basin is caused, and the anomalous southerly (easterly) airflow and water vapor convergence (divergence) maintain in the basin, which is prone (not prone) to the low-latitude warm and humid air transporting to the basin, resulting in the increase (decrease) of water vapor inflow at the southern boundary of the basin and more (less) summer precipitation in the basin.

In generally, in the context of global climate change, the water vapor inflow at the western and southern boundaries of Sichuan Basin has decreased significantly in the past 38 years, and the summer precipitation in the basin also has a decreasing trend. The decrease in net water vapor budget in the Sichuan basin has led to a shortage of water resources, which has affected all aspects of social economy and people's lives in Sichuan. Under the influence of global warming, accelerated urbanization, and abnormal atmospheric circulation, the reduction in water vapor budget in the basin may be more obvious in the future, and so the water resources problem in the Sichuan Basin deserves more attention.

**Author Contributions:** Conceptualization, Y.L.; Methodology, C.Z.; writing—original draft preparation, D.Q. All authors have read and agreed to the published version of the manuscript.

**Funding:** This research was jointly funded by the Second Tibetan Plateau Scientific Expedition and Research (STEP) program (Grant No.2019QZKK0103), the National Natural Science Foundation of China (Grant No.91937301), the Scientific and Technological Research Program of China Railway Eryuan Engineering Group CO. LTD (Grant No. KYY2020066(20–22)), the co-sponsored project by Sichuan Meteorological Bureau and Nanjing University of Information Science & Technology (SCJXHZ04), and Heavy Rain and Drought-Flood Disasters in Plateau and Basin Key Laboratory of Sichuan Province (SCQXKJQN2019012).

**Institutional Review Board Statement:** Not applicable.

**Informed Consent Statement:** Not applicable.

**Data Availability Statement:** The daily precipitation data used in the study were obtained from the China Meteorological Administration Information Center. The monthly average ERA-Interim reanalysis data are available through the ECMWF, https://apps.ecmwf.int/datasets/ (accessed on 15 September 2018).

**Acknowledgments:** We thank the two anonymous reviewers for their time and support in reshaping the manuscript. Their comments helped us to improve the final version of the manuscript. Special appreciation to the China National Meteorological Information Center and ECMWF, for the provision of the datasets used in the study.

**Conflicts of Interest:** The authors declare no conflict of interest.

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
