# Peer review of "Variation Characteristics of Summer Water Vapor Budget and Its Relationship with the Precipitation over the Sichuan Basin"

_water, doi:10.3390/w13182533_

Round 1

Reviewer 1 Report

Manuscript ID: water-1358526

The manuscript examines the summer water vapor variability and its relation with the precipitation over the Sichuan Basin. The methodology is adequately designed, and the authors have explained the results as shown in the figures and tables. However, I do see that the manuscript essentially repeats the analyses done by others over a somewhat larger region comprising the Sichuan basin. Therefore, I do not find the results surprising or novel. The manuscript has several grammatical mistakes, and it also needs a proper revision to add a few clarifications. After carefully examining the manuscript, I recommend accepting the manuscript after a minor revision.

Major comments:

  1. The manuscript has several grammatical mistakes and typos. Please revise the text to remove any grammatical errors and typos.
  2. The authors have discussed many previous research in the introduction. However, the introduction needs to be revised to build a nice storyline and a motivation for the current study. Because some previous studies have done the same analysis over a region encompassing the Sichuan province/ basin, the authors need to highlight the novelty aspect of their study in a better way.

Minor comments:

  1. Line 92: “decreasing”.
  2. Line 96-97 is somewhat a repetition of lines 81-83.
  3. Lines 124-128: Grammatically incorrect and confusing. Please rewrite the text.
  4. Please indicate that the negative budget indicates outflow and the positive budget shows inflow.
  5. Fig. 1: Please indicate that the boundary shows Sichuan province and the rectangles outline the Sichuan Basin. Also, please show the topography to highlight the elevation levels. Amend the text throughout the manuscript accordingly.
  6. At several places, the authors have discussed “significantly” increasing and/ or decreasing trends. Please indicate the level of significance (e.g., 5% or 10%). Please revise the manuscript to add the significance level wherever necessary.
  7. Fig. 2: For clarity, please show positive and negative trends with different line types (e.g., solid and dashed). Also, since each budget is indicated by a different color, use solid curves only for simplicity.
  8. Line 259: Replace “Seen from” with “As shown in”.
  9. Line 260: summer precipitation in Sichuan (add “province” here for clarity) is mainly over the Sichuan Basin.
  10. Lines 259-266: Show the locations of these stations in Fig. 3.
  11. Line 262: Replace “large-value centers of precipitation” with “centers of precipitation maxima.”
  12. Lines 267-268: Please correct “large-value center” as indicated in the previous point.
  13. Fig. 3: The caption should be revised to indicate that the figure shows mean (or total) precipitation during the summer season over the period 1979-2016.
  14. Lines 280-288: Are the trends discussed statistically significant? If yes, please indicate that.
  15. Line 298: Start the sentence with “As shown in”.
  16. Figure 5: Replace “shadow” with “shaded”.
  17. Fig. 2a: The caption should read “Net water vapor budget”.

Author Response

Dear reviewer:

Firstly, we would like to thank you for your constructive comments and suggestions concerning our manuscript entitled “Variation Characteristics of Summer Water Vapor Budget and its Relationship with the Precipitation over the Sichuan Basin” (ID: water-1358526). These comments and suggestions are all valuable and helpful for improving our article, as well as the important guiding significance to our researches. 

All the authors have seriously discussed about all the comments. We have revised the manuscript and careful consideration is given to the points raised by the reviewers. The changes in the manuscript are highlighted in red font. These changes will not influence the content and framework of the paper. Point-by-point responses to the reviewers are listed below this letter.

Once again, thank you very much for your comments and suggestions. We hope that the revision is acceptable and look forward to hearing from you soon.

With best regards,

Yours sincerely,

Dongmei Qi, YueQing Li, Changyan Zhou

Response to Reviewer 1 Comments

Major comments:

1. The manuscript has several grammatical mistakes and typos. Please revise the text to remove any grammatical errors and typos.

Response 1: Thank you for your reminding. We have revised the text. Please see P1, Line 11 deleted “the”. P2, Line 91. P3, Lines 124,125,127, 135-138. P6, Lines 231, 243, 262-263, 267-269. P7, Line 272. P8, Lines 295-296, 298-299, 304-305. P9, Line 333. P12, Lines 431, 454. P13, Lines 491-494, 506. P14, Line 520.

2. The authors have discussed many previous research in the introduction. However, the introduction needs to be revised to build a nice storyline and a motivation for the current study. Because some previous studies have done the same analysis over a region encompassing the Sichuan province/ basin, the authors need to highlight the novelty aspect of their study in a better way.

Response 2: Special thanks for constructive and helpful suggestion. We have revised the introduction. Please see P3, Lines 111-123.

Minor comments:

  1. Line 92: “decreasing”.

Response 1: Special thanks for your careful reminding. We have reasonably changed “decrease” to “decreasing” in P2, Line 91.

2.Line 96-97 is somewhat a repetition of lines 81-83.

Response 2: Special thanks for your careful reminding. The original statements have been checked and the repeated sentences have been adjusted. Please see P2, Lines 81-82 in the revised manuscript.

3.Lines 124-128: Grammatically incorrect and confusing. Please rewrite the text.

Response 3: Following the Reviewer’s suggestion, we have rewrote the text. Please see P3, Lines 135-138.

4.Please indicate that the negative budget indicates outflow and the positive budget shows inflow.

Response 4: Following the Reviewer’s suggestion, we have added relevant descriptions. Please see P4, Lines 165-169.

5.Fig. 1: Please indicate that the boundary shows Sichuan province and the rectangles outline the Sichuan Basin. Also, please show the topography to highlight the elevation levels. Amend the text throughout the manuscript accordingly.

Response 5: Following the Reviewer’s suggestion, we have added topographic features in the Fig. 1, and modified the text. Please see P5, Lines 193-199.

6. At several places, the authors have discussed “significantly” increasing and/ or decreasing trends. Please indicate the level of significance (e.g., 5% or 10%). Please revise the manuscript to add the significance level wherever necessary.

Response 6: Special thanks for your careful reminding. We have revised the manuscript. Please see P6, Lines 230-270. P7, Lines 271-279. P8, Lines 317, 322.

7. Fig. 2: For clarity, please show positive and negative trends with different line types (e.g., solid and dashed). Also, since each budget is indicated by a different color, use solid curves only for simplicity.

Response 7: Special thanks for constructive and helpful suggestion. We have modified the figure 2 and title. Please see P7, Line 280-287.

8. Line 259: Replace “Seen from” with “As shown in”.

Response 8: Special thanks for your careful reminding. We have reasonably changed “Seen from” to “As shown in” in P8, Line 295.

9. Line 260: summer precipitation in Sichuan (add “province” here for clarity) is mainly over the Sichuan Basin.

Response 9: Following the reviewer’s suggestion, we have added province in P8, Line 296 in the revised manuscript.

10. Lines 259-266: Show the locations of these stations in Fig. 3.

Response 10: Following the reviewer’s suggestion, we have revised Figure 3, and showed the locations of Wanyuan, Ya'an and Beichuan stations. Please see P8, Line 311.

11. Line 262: Replace “large-value centers of precipitation” with “centers of precipitation maxima.”

Response 11:  Special thanks for your careful reminding. We have reasonably changed “large-value centers of precipitation” to “centers of precipitation maxima” in P8, Lines 298-299.

12. Lines 267-268: Please correct “large-value center” as indicated in the previous point.

Response 12: Following the reviewer’s suggestion, we have corrected the sentence, Please see P8, Lines 304-305 in the revised manuscript.

13. Fig. 3: The caption should be revised to indicate that the figure shows mean (or total) precipitation during the summer season over the period 1979-2016.

Response 13: Thank you for your reminding. We have revised the caption. Please see P8, Line 311.

14. Lines 280-288: Are the trends discussed statistically significant? If yes, please indicate that.

Response 14: Thank you for your reminding. We have indicated statistically significant. Please see P8, Lines 317, 322.

15. Line 298: Start the sentence with “As shown in”.

Response 15: Following the reviewer’s suggestion, we have revised the sentence, Please see P9, Line 333 in the revised manuscript.

16. Figure 5: Replace “shadow” with “shaded”.

Response 16: Following the reviewer’s suggestion, we have reasonably changed “shadow” to “shaded” in P12, Line 454 in the revised manuscript.

17. Fig. 2a: The caption should read “Net water vapor budget”.

Response 17: Special thanks for your careful reminding. We have revised the caption. Please see P7, Line 283 in the revised manuscript.

ʉ۬

Author Response

Dear reviewer:

Firstly, we would like to thank you for your constructive comments and suggestions concerning our manuscript entitled “Variation Characteristics of Summer Water Vapor Budget and its Relationship with the Precipitation over the Sichuan Basin” (ID: water-1358526). These comments and suggestions are all valuable and helpful for improving our article, as well as the important guiding significance to our researches. 

All the authors have seriously discussed about all the comments. We have revised the manuscript and careful consideration is given to the points raised by the reviewers. The changes in the manuscript are highlighted in red font. These changes will not influence the content and framework of the paper. Point-by-point responses to the reviewers are listed below this letter.

Once again, thank you very much for your comments and suggestions. We hope that the revision is acceptable and look forward to hearing from you soon.

With best regards,

Yours sincerely,

Dongmei Qi, YueQing Li, Changyan Zhou

Response to Reviewer 2 Comments

The manuscript is confusing. It should be organized in introduction, material and methods and results/discussion, conclusions sections.

Response 1: Following the reviewer’s suggestion, we have reorganized the sections, sections 1. Introduction, 2. Material and methods, 3. Results, and 4. Conclusions sections. Please see P3, Line 134. P5, Line 201. P13, Line 490.

In the introduction, the description of the work should be improved.

Response 2: Following the reviewer’s suggestion, we have revised the introduction. Please see P2, Lines 51-52. P3, Lines 111-123.

Introduction: please emphasize the novelty of the study.

Response 3: Thank you for your reminding, we have emphasized the novelty of the study. Please see P3, Lines 111-123.

Lines 51, 52: “The increasing frequency of flood events from 1961 to 2005 is mainly caused by the increase of water vapor transport”.

Have the authors investigated all the possible causes? If not, please rephrase.

For instance is some catchements, the increase in the frequency of occurrences of flood events is related to non-stationary features of an average catchment scale rainfall-runoff erosivity index.

Please see:

Longobardi, Antonia, Nazzareno Diodato, and Mirka Mobilia. "Historical storminess and hydro- geological hazard temporal evolution in the solofrana river basin—Southern Italy." Water 8.9 (2016): 398.

Response 4: Thank you for your reminding, we have rephrased the sentence. Please see P2, Lines 51-52.

Lines 124, 125: please rephrase

Response 5: Following the Reviewer’s suggestion, we have rephrased the text. Please see P3, Lines 135-138.

Equ. 1 and 2: please define u, v, ,

Response 6: Following the reviewer’s suggestion, we have defined u, v, , , Please see P4, Line 178. “u is the meridional wind, v is the zonal wind, is longitude, is latitude.”

Table l: please define Bt, Bn etc.

Response 7: Following the Reviewer’s suggestion, we have added relevant descriptions. Please see P4, Lines 165-169.

Line 207: rephrase

Response 8: Thank you for your reminding, we have rephrased the sentences. Please see P6, Line 231.

Figure 3: is the summer precipitation a long-term average precipitation? Please define the sorce of the data.

Response 9: Yes, it is. Thank you for your reminding. We have revised the caption. Please see P8, Line 311.

Please split the chapter 5. It is too long and hard to read

Response 10: Special thanks for your careful reminding. Adjustments have been made to improve the structure of the text. Please see P9, Line 332. P11, Line 386.

Line 351: How have the wet and drought summers been selected?

Response 11: We select summer precipitation greater than 650mm as wet summer, and summer precipitation less than 450mm as drought summers. We have revised the text. Please see P11 Lines 387-388 in the revised manuscript.

Conclusion: Which suggestions do the results give to the readers?

Response 12: Following the reviewer’s suggestion, we have added suggestions at the end of the conclusion. Please see P14, Lines 548-556.
